# Midline Gliomas: A Retrospective Study from a Cancer Center in the Middle East

**DOI:** 10.3390/cancers15184545

**Published:** 2023-09-13

**Authors:** Sarah Al Sharie, Dima Abu Laban, Jamil Nazzal, Shahad Iqneibi, Sura Ghnaimat, Abdallah Al-Ani, Maysa Al-Hussaini

**Affiliations:** 1Faculty of Medicine, Yarmouk University, Irbid 21163, Jordan; sarahalsharie2000@gmail.com; 2Department of Radiology, King Hussein Cancer Center, Amman 11941, Jordan; da.11945@khcc.jo; 3Office of Scientific Affairs and Research, King Hussein Cancer Center, Amman 11941, Jordan; jamil.nazzal@outlook.com (J.N.); shahad.iqneibi@yahoo.com (S.I.); sa.12608@khcc.jo (S.G.); abdallahalany@gmail.com (A.A.-A.); 4Department of Pathology and Laboratory Medicine, King Hussein Cancer Center, Amman 11941, Jordan

**Keywords:** diffuse midline glioma (DMG), circumscribed glioma, H3 K27me3 alteration, H3 K27M mutation, corpus callosum, brainstem, thalamus, spinal cord

## Abstract

**Simple Summary:**

Midline gliomas are tumors that can manifest as either circumscribed or diffuse growths. These tumors typically develop in midline structures, including the thalamus, brainstem, and spinal cord. Other structures along the midline, such as the corpus callosum, basal ganglia, ventricles, paraventricular structures, and cerebellum may also be affected. Diffuse midline gliomas (DMGs) are a subtype of midline gliomas and are considered highly aggressive and are most commonly diagnosed in children and young adults. These tumors are difficult to treat due to their location and invasive nature, and their prognosis is generally poor. We aim to describe our experience with midline gliomas, with a focus on DMGs.

**Abstract:**

Midline gliomas are tumors that occur in midline structures and can be circumscribed or diffuse. Classical midline structures include the thalamus, brainstem, and spinal cord. Other midline structures include the corpus callosum, basal ganglia, ventricles, paraventricular structures, and cerebellum. Diffuse midline glioma (DMG) is a diffuse glioma that occurs in the classical midline structures, characterized by a specific genetic alteration, and associated with grim outcome. This study was conducted at King Hussein Cancer Center and reviewed the medical records of 104 patients with circumscribed and diffuse gliomas involving midline structures that underwent biopsy between 2005 and 2022. We included a final cohort of 104 patients characterized by a median age of 23 years and a male-to-female ratio of 1.59-to-1. Diffuse high-grade glioma (DHGG) was the most common pathological variant (41.4%), followed by DMG (28.9%). GFAP was positive in most cases (71.2%). Common positive mutations/alterations detected by surrogate immunostains included H3 K27me3 (28.9%), p53 (25.0%), and H3 K27M (20.2%). Age group, type of treatment, and immunohistochemistry were significantly associated with both the location of the tumor and tumor variant (all; *p* < 0.05). DMGs were predominantly found in the thalamus, whereas circumscribed gliomas were most commonly observed in the spinal cord. None of the diffuse gliomas outside the classical location, or circumscribed gliomas harbored the defining DMG mutations. The median overall survival (OS) for the entire cohort was 10.6 months. Only the tumor variant (i.e., circumscribed gliomas) and radiotherapy were independent prognosticators on multivariate analysis.

## 1. Introduction

Primary central nervous system (CNS) tumors comprise 2–3% of malignant tumors, with gliomas being the most common primary intracranial neoplasm [1]. Per the 2016 WHO classification of CNS tumors, diffuse midline gliomas (DMGs) were presented as a new CNS tumor entity that is primarily characterized by diffuse growths within midline structures, namely the thalamus, brainstem, and spinal cord [2]. Moreover, DMGs were thought to only harbor H3 K27M mutations within either the histone H3 H3F3A or the HIST1H3B/C genes; thus, the H3 K27M—mutant variant was defined. The second update of the Consortium to Inform Molecular and Practical Approaches to CNS Tumor Taxonomy (cIMPACT-NOW) in 2018 recommended that the DMG, H3 K27M—mutant label should only be applied for tumors that are diffuse, midline, gliomas, and harbor an H3 K27M mutation [3]. The consortium also acknowledges the presence of H3 K27M mutations within a number of other CNS tumors. Other gene mutations including p53, ACVR1, ATRX, and PI3K have been associated with DMG [4,5,6,7]. Subsequently, based on the latest WHO CNS tumors classification in 2021, these tumors are now described as H3 K27-altered gliomas, CNS WHO grade 4 [8]. 

DMG is mainly a pediatric tumor with no gender predilection [9]. It is the 2nd most common malignant childhood tumor in the United States, accounting for nearly two-thirds of pediatric brainstem tumors and nearly 20% of all pediatric CNS tumors [9]. The greater majority of DMG occurs between 5 and 10 years of age and is associated with a median overall survival of 9–12 months post-diagnosis [9,10]. It can, however, present in adults with a reported average age of 30 years at the time of diagnosis [11,12]. DMG is known for its aggressive nature, dismal outcomes, and poor prognosis with a 99% 5-year mortality rate [13]. Due to the high morbidity associated with surgery in such an eloquent region of the brain and sampling errors associated with biopsies, the treatment options for DMG are confined to either chemotherapy, radiotherapy, or targeted molecular agents, all of which are associated with minimal effectiveness [14,15]. 

Interestingly, the genetic alteration profile of DMG was reported in a number of other CNS tumors. H3 K27M mutations have been described in circumscribed tumors, such as pilocytic astrocytoma (PA) and ganglioglioma (GG), occurring in midline structures, with a debatable effect on prognosis [16]. In addition, H3 K27-altered diffuse gliomas have been described in other midline locations outside the typical locations including the corpus callosum [17]. 

King Hussein Cancer Center (KHCC) is the only stand-alone comprehensive cancer center in Jordan and among the very few in the Middle East, serving approximately 60% of all cancer cases in Jordan [18]. It treats both adult and pediatric cases. Per the KHCC’s 2018 Cancer Registry report, CNS tumors constituted the 11th most common tumors at the center with a total of 216 cases (4.5% of all diagnosed cases) stratified into 154 adult and 62 pediatric cases.

In this study, we aimed to describe the clinicopathological characteristics and outcomes of patients with midline gliomas, primarily focusing on DMGs in the classical locations (thalamus, brainstem, and spinal cord). In addition, per the recommendation of the cIMPACT-NOW, we investigated the staining patterns for H3 K27M in conjunction with H3 K27me3 in circumscribed gliomas, including PA and GG, occurring in midline structures. Other diffuse gliomas in other more recently described midline locations, such as the corpus callosum and basal ganglia, were also investigated.

## 2. Materials and Methods

### 2.1. Design, Setting, and Participants

We performed a retrospective chart review of all gliomas involving midline structures (i.e., thalamus, brainstem, spinal cord, corpus callosum, basal ganglia, parasagittal gurus, intraventricular areas, and cerebellum) that underwent biopsy/resection and were diagnosed and treated at KHCC between 2005 and 2022.

Clinical data collected from electronic medical records included age (pediatrics, <18 years; adults, >18 years), biological sex, tumor location, radiological features (i.e., circumscribed vs. infiltrative), treatment offered (i.e., chemotherapy, radiotherapy), and dates of last available contact or death. Among our cohort, surgical treatment manifested in the form of diagnostic biopsy due to the associated morbidity with surgical resection with the delicate affected areas. In terms of administered treatment regimens, chemotherapy consisted primarily of Temozomolide, most commonly as the only agent, with occasional cases receiving Lomustine in the pediatric age group. Vincristine with carboplatin was the second most common regimen received, mostly in the pediatric age group. Radiotherapy was received mostly at a dose of 5400 Gys. Patients diagnosed with pontine gliomas were treated with focal radiotherapy with or without Prednisolone. However, patients diagnosed with DMG at other locations including thalamic DMGs were treated with focal radiotherapy with daily Temozolomide. Overall survival (OS) was defined as the time between the date of radiological diagnosis to the date of last available follow-up or death due to any cause. Survival status was supplemented with data from the National Civil Status and Passports Department in Jordan. 

This study was approved by the Institutional Review Board (IRB) of King Hussein Cancer Center, Amman, Jordan; IRB Reference: 21 KHCC 171. The IRB follows the guidelines of Good Clinical Practice (GCP) and the Declaration of Helsinki.

### 2.2. Pathology and Immunohistochemistry

Pathology reports were reviewed and the pathological data collected included the pathological diagnosis at the time of authorizing the pathological report in addition to the grade; the latter only applies to cases issued before 2016. The diagnosis of DMG was rendered only to more recent cases following the release of the 2016 WHO classification. Immunohistochemistry was performed on a BenchMark Ultra immunostainer (Ventana Medical Systems, Tucson, AZ, USA). Slides were pretreated with Cell Conditioning Solution CC1 (Ventana Medical Systems) for 32 min at room temperature. Primary antibodies were incubated at 37 °C for 32 min, followed by Ventana standard signal amplification, UltraWash, counter-staining with one drop of hematoxylin for 4 min, and one drop of bluing reagent for 4 min. UltraView Universal DAB Detection Kit (Ventana Medical Systems) was used for visualization. Immunostains for GFAP (1:2000, Cell signaling, Promega), p53 (clone DO-7, Dako), H3 K27M (oncohistone mutant Antibody, 100 µL, thermofisher), H3 K27me3 (Polyclonal Antibody, 100 µL, thermofisher), ATRX (1:2000, BSB3296, BioSB, Santa Barbara, CA, USA), and IDH-1 (p.R132H) (1:2, clone 1) were performed on all cases with available paraffin blocks and adequate material. Positive and negative controls were run for each antibody as appropriate. 

GFAP was considered positive if >10% of the tumor showed positive staining. p53 was considered to be mutant when >50% of the tumor cells showed strong nuclear staining. H3 K27M was considered mutant if >90% of tumor cells showed strong nuclear staining in the presence of negative internal control (endothelium and neurons), whereas H3 K27me3 was considered altered when the tumor cells showed attenuated to negative nuclear staining in the presence of retained nuclear staining in internal control (endothelium and neurons). ATRX was considered mutant if the tumor cells showed loss of nuclear stain in the presence of internal positive control (endothelium and neurons). IDH-1 (p.R132H) immunostain was considered mutant if the tumor cells showed positive cytoplasmic, and to a lesser extent, nuclear staining. 

### 2.3. Statistical Analysis

All data analyses were conducted using Stata version 17 software (StataCorp. 2021. Stata: Release 17. Statistical Software. StataCorp LLC., College Station, TX, USA). Normality of variables was assessed using histograms or quantile–quantile plots in addition to the Kolmogorov–Smirnov and Shapiro–Wilk tests. Normally distributed continuous variables were presented as means ± standard deviations. Conversely, asymmetrical continuous variables were presented as medians with interquartile ranges (IQR). Categorical variables were presented as frequencies with associated percentages [n (%)].

Patients’ clinicopathological characteristics were compared per tumor location and pathological entities (i.e., DMG, diffuse low-grade gliomas (DLGG), diffuse high-grade glioma (DHGG), circumscribed gliomas (CG)). Associations between categorical variables were assessed by the chi-square, whereas associations involving continuous variables were examined using the Mann–Whitney U test and the analysis of variance test (ANOVA) as appropriate. OS was plotted per different clinicopathological characteristics using the Kaplan–Meier method. Differences in overall survival were examined using the log-rank test. Multivariate cox regression was employed to detect predictors of overall survival. Included predictors were either extracted from the relevant literature or were significantly associated with survival on univariate analysis. A p-value of less than 0.05 was considered statistically significant for all conducted analyses.

## 3. Results

### 3.1. Characteristics of Included Participants

The final cohort consisted of 104 patients with adequate tissue samples. The mean and median ages of included participants were 29.4 (range, 1–74) and 23 years (IQR, 10–48), respectively. The greater majority of cases were adults (58.3%) compared to their pediatric counterparts (41.7%). Similarly, males comprised 61.5% of total cases compared to 38.5% by females. In terms of pathological variants, DHGG was the most common among our patients (41.4%), followed by DMG (28.9%) and CG (20.2%). GFAP was positive in most cases (71.2%). Common positive mutations/alterations detected by surrogate immunostains included H3 K27me3 (28.9%), p53 (25.0%), and H3 K27M (20.2%). In terms of treatment, a greater proportion of patients received radiotherapy (51.9%), whereas 42.3% were started on chemotherapy. Overall, 43.3% of participants died, 31.7% were lost to follow-up, and only 25.0% were alive at the time of data collection.

### 3.2. Univariate Analysis

On univariate analysis, the location of the tumor was significantly associated with age group (*p* = 0.001), pathological variants (*p* < 0.05), immunohistochemistry (*p* < 0.05), and type of received intervention (*p* < 0.05). 

When mapped per tumor location variant, the greater majority of thalamic tumors were of the DMG subtype (54.8%), followed by DHGG (25.8%). CGs and DMGs were the most prevalent within the brainstem (38.5% and 38.5%, respectively) and spinal cord (43.5% and 34.8%, respectively). As for the basal ganglia and corpus callosum, DHGG tumors were the most prevalent (40.0% and 95.5%, respectively). On another note, outcome was not significantly associated with the location of tumors (*p* = 0.208). Table 1 summarizes the demographics, clinical, and immunohistochemical characteristics stratified by the primary site of the tumor. Of note, none of the DHGG or DLGG outside the classical locations for DMG exhibited H3 K27M or H3 K27me3 alterations. 

When stratified per pathological variant, the oldest group was DHGG cases (46 years [2–74], whereas the youngest variant of midline tumors was CGs (8 years [1–48]) (Table 2). Age group (*p* = 0.001), locations other than brainstem (*p* < 0.05), immunohistochemistry (*p* < 0.001), radiotherapy (*p* = 0.004), and outcome (*p* = 0.001) were significantly associated with pathological variants. In terms of immunohistochemistry, all but one stain (i.e., IDH1) was observed at the highest frequency among patients with DMG. The H3 K27M mutation or H3 K27me3 alterations were only observed among DMG cases. None of the circumscribed gliomas were stained with either H3 K27M or showed loss of the H3 K27me3. Mortality was highest for DMG (60%), followed by DHGG (53.5%), and lowest for CGs (4.76%). Figure 1 demonstrates a schematic summary of clinical and immunohistochemical characteristics of the 104 included cases of midline gliomas.

### 3.3. Survival Analysis

For all included participants, the median OS was 10.6 months. Pediatrics had a median OS of 85.2 months compared to 23.9 months for adults; survival differences were insignificant between age groups (*p* = 0.268) (Figure 2A). Similarly, median OS differences between males (41.3 months) and females (29.4 months) were statistically insignificant (*p* = 0.369) (Figure 2B). There was a statistically significant difference in OS among pathological entities (*p* < 0.001) (Figure 2C). Cases with DHGG had a median OS of 10.0 months whereas those with DMG had a median OS of 18.1 months. When stratified by location, the median OS did not significantly differ among different locations (*p* = 0.173) (Figure 2D). The median OS for locations is as follows in descending order: spinal cord (85.2 months), corpus callosum (23.9 months), thalamus (20.3 months), and brainstem (18.1 months). 

Figure 3 shows survival differences stratified per immunohistochemical stains including p53 (Figure 3A), H3 K27M (Figure 3B), H3 K27me3 (Figure 3C), and ATRX (Figure 3.D). OS survival was significantly higher for non-mutated H3 K27M variants (*p* = 0.041) (Figure 3B) and positive H3 K27me3 variants (*p* = 0.015) (Figure 3D). 

Treatment modality did not impact OS among included patients (Figure 4A,B).

### 3.4. Multivariate Survival Analysis

Table 3 demonstrates the predictors of survival among included participants. Compared to patients with DHGG, cases of CGs were associated with a significantly lower risk of mortality (HR 0.013, CI 95%: 0.001–0.142, *p* < 0.001). Additionally, radiotherapy was the only treatment modality that affected survival outcomes (HR 0.298, CI 95%: 0.090–0.987, *p* = 0.048).

### 3.5. Pathological Examples of DMG in Various Locations

Figure 5, Figure 6 and Figure 7 show examples of DMG cases occurring in various locations, including a case in the thalamus, the pons, and the spinal cord. The diagnosis was based in all cases on the combination of mutant H3 K27M and altered H3 H27me3 immunostains. 

## 4. Discussion

In this paper, we examined the clinicopathological characteristics of 104 patients with midline gliomas. Only tumors fulfilling the WHO definition of DMG demonstrated the characteristic staining pattern, whereas tumors in other locations, or circumscribed tumors, failed to stain similarly to DMGs. When categorized by histological diagnosis and grade, DHGG was the most common variant among our included cohort. There were significant associations between tumor location and a number of variables including age group, pathological variant, immunostaining, and therapy. Similarly, pathological variants were significantly associated with age group, immunostaining, outcome, and certain locations. Pathological entity and radiotherapy administration were the only predictors of survival among our patients. 

DMG is considered the most prevalent pediatric malignant brainstem neoplasm with 200–300 occurrences per year in the United States [19]. Also, DMGs makeup approximately two-thirds of pediatric brainstem neoplasms and often develop in children between the ages of 3 and 10 years old [20], compared to approximately 4% in adults [21,22]. Recently, however, the incidence of radiologically defined DMGs in the adult population made up 19% of all adult gliomas [23]. Among our included cases, the greater majority of patients with DMG were adults. In terms of survival, DMGs are universally deadly with only a few exceptions and limited treatment options [24]. The reported median OS ranged from 9 to 11 months [25,26]. Within our cohort, the median OS for DMG cases was higher than the reported literature at 18.1 months. It should be noted that discrepancies between our experience with patients with midline gliomas and the literature might be attributed to our case selection process. Only patients with biopsies were included, whereas cases in which treatment was started based on only radio imaging were excluded.

Zheng et al. studied 164 patients with H3 K27M-mutant DMGs [27]. Similar to our findings, adults were more represented, with an average age of 24 years. Across their cohort, the most common tumor locations were the brainstem in children and the thalamus in adults. Furthermore, their immunohistochemistry analysis showed positive staining for H3 K27M mutation in most cases. However, in contrast to our findings, their survival analysis demonstrated that age (i.e., older adults) and loss of ATRX expression were associated with better survival. Such discrepancies may be attributed to sample size differences, which could have limited the statistical power of our conducted analyses.

Recently, DMGs have been described outside their classical locations. In a study involving 47 cases of H3 K27M-mutant DMGs, their most common locations were the pons (36.2%), followed by the thalamus (31.9%) and spinal cord (19.1%). Additionally, DMG cases residing within the third ventricle (6.2%), hypothalamus (2.1%), cerebellum (2.1%), and pineal gland (2.1%) were also reported across the literature [6,28]. Wang et al. reported five cases of H3 K27M-mutant DMGs in the corpus callosum, a structure that is not considered “midline” per the WHO classification for “DMG, H3 K27M-mutant” tumors [29]. Such cases were associated with poorer prognosis compared to DMGs of classical locations. In our study, most DMG cases were located in the thalamus, whereas the remaining occurred in the spinal cord and brainstem. Despite their relatively large number, none of our cases in the corpus callosum or other locations showed the characteristic H3 K27M mutation or H3K27me3 alteration. 

Another interesting observation is the reporting of H3 K27M mutation among CGs such as PA and GG occurring within midline structures [30,31]. The effect on outcome was inconsistently reported in the literature; some of these tumors were noticed to have a good survival rate, whereas the others had a very poor prognosis [31,32,33]. A study by Li et al. reported that H3 K27M-mutant CGs had significantly worse prognoses compared to *H3* K27M-wildtype CGs [34]. In our study, none of the CG cases were found to have an H3 K27M mutation. Moreover, those cases were associated with improved overall survival compared with DHGG.

Until the present time, the only treatment that increases life expectancy in patients with DMGs is radiation therapy. The standard radiotherapy regimen is 54–60 Gy over a 6-week period and was found to reduce the progression of the tumor in 70–80% of patients [35,36]. On multivariate analysis, only radiotherapy significantly predicted survival, which is in concordance with the established literature. It should be noted that efforts to develop DMG-centric treatments have been motivated by recent developments in the understanding of the molecular biology behind H3 K27M-mutant DMGs. Several methods have been proposed to manage DMGs including tumor-localized therapy (e.g., Convection-Enhanced Delivery) and immunotherapy [10]. Additionally, targeting different genes responsible for tumorigenesis (e.g., H3 K27M, ACVR1, and EZH2) could result in novel and effective treatment methods that may positively impact survival outcomes [15,37,38].

Furthermore, targeted molecular therapy has emerged as a promising approach to the management of DMGs [39]. As of 2023, there are no FDA-approved targeted molecular therapies for DMGs [40]. However, there are several targeted therapies that are currently in clinical trials for DMG including ONC201, also known as TIC10, which is an orally available small molecule. It works by enhancing the action of a TNF-related apoptosis-inducing ligand (TRAIL) protein, which triggers programmed cell death in cancer cells. The restoration of cells’ self-destruction capability is believed to inhibit tumor growth. Remarkably, ONC201 can breach the blood–brain barrier, suggesting that it could be a promising treatment modality for brain tumors, an area where traditional chemotherapy faces limitations [41,42]. Moreover, PI3K inhibitors, the likes of Paxalisib (Vitrakvi) and GDC-0084, are also undergoing clinical trials for the treatment of chemotherapy-resistant DMGs [43].

We acknowledge a number of limitations in our study. Firstly, the retrospective nature of this study. Secondly, all patients were recruited from a single center. However, KHCC is Jordan’s most comprehensive and advanced cancer center; thus, it is expected that most, if not all, cases of high-grade tumors will be managed at the center. Thirdly, only patients who underwent biopsy were included in this study, which led to the exclusion of patients diagnosed only on a radiological basis. Fourthly, our cohort lacked molecular classification. Fifthly, follow-up and survival data were not available for a large number of cases, which could have limited our pool of included cases and therefore, our statistical power when conducting survival analyses. Other limitations include the relatively small sample size and the short follow-up period. The latter is expected as not all patients are able to cover the costs of management and follow-up following initial treatment.

## 5. Conclusions

In conclusion, we report our findings of midline gliomas, both diffuse and circumscribed. In general, our findings are consistent with international literature. None of our diffuse glioma cases outside the classical locations fulfilled the criteria for DMG. Also, none of the CGs stained for H3 K27M nor showed H3K27me3 alterations. Finally, administration of radiotherapy was the only treatment-related predictor of survival.

## Figures and Tables

**Figure 1 cancers-15-04545-f001:**
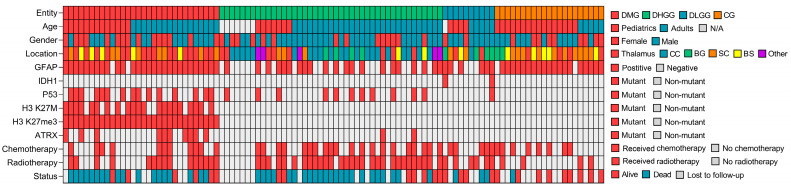
Schematic summary of clinical and molecular characteristics of 104 midline gliomas. DMG: Diffuse Midline Glioma, DLGG: Diffuse Low-Grade Glioma, DHGG: Diffuse High-Grade Glioma, CG: Circumscribed Glioma, BS: Brainstem, BG: Basal Ganglia, CC: Corpus Callosum, SC: Spinal Cord, N/A: Not Applicable.

**Figure 2 cancers-15-04545-f002:**
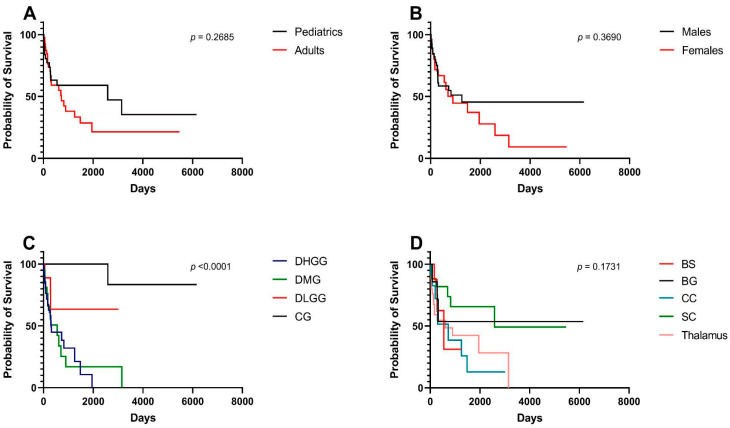
Kaplan–Meier survival curves and log-rank tests for midline gliomas and associated clinical factors. (**A**) Pediatrics (age < 18) vs. adults (age > 18). (**B**) Males vs. females. (**C**) Pathological grades—DHGG: Diffuse High-Grade Glioma, DMG: Diffuse Midline Glioma, DLGG: Diffuse Low-Grade Glioma, CG: Circumscribed Glioma. (**D**) Tumor locations, BS: Brainstem, BG: Basal Ganglia, CC: Corpus Callosum, SC: Spinal Cord.

**Figure 3 cancers-15-04545-f003:**
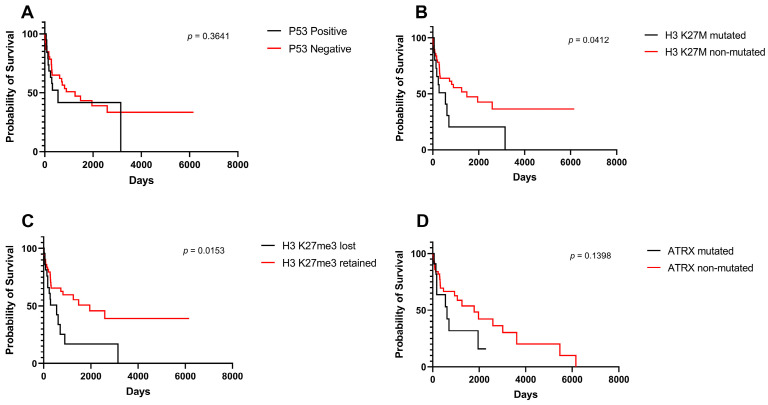
Kaplan–Meier survival curves and log-rank tests for midline gliomas based on surrogate immunohistochemical markers. (**A**) p53 mutation, (**B**) H3 K27M mutation, (**C**) H3 K27me3 alteration, (**D**) ATRX mutation.

**Figure 4 cancers-15-04545-f004:**
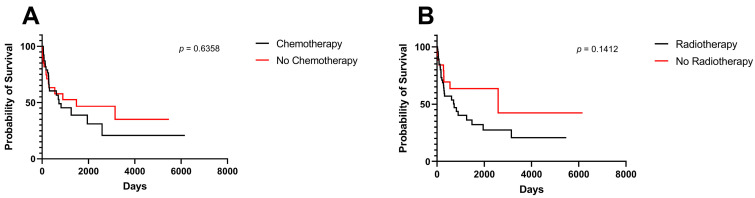
Kaplan–Meier survival curves and log-rank tests for midline gliomas. (**A**) Chemotherapy, (**B**) radiotherapy.

**Figure 5 cancers-15-04545-f005:**
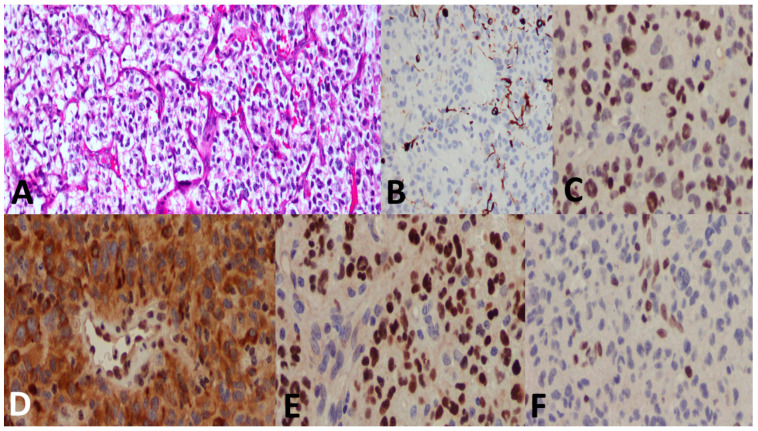
A 27-year-old male patient with a thalamic tumor. (**A**) A highly cellular tumor with an oligodendroglioma-like component and microvascular proliferation. (**B**) GFAP immunostain was at large negative. (**C**) p53 showed positive nuclear staining in most of the tumor cells. (**D**) ATRX immunostain shows loss of nuclear stain (endothelium serves as a positive internal control). (**E**) H3 K27M shows positive nuclear stain (endothelium serves as the negative internal control). (**F**) H3 K27me3 shows loss of nuclear stain (endothelium serves as the positive internal control). The final diagnosis was DMG, H3 K27-altered, CNS WHO grade 4.

**Figure 6 cancers-15-04545-f006:**
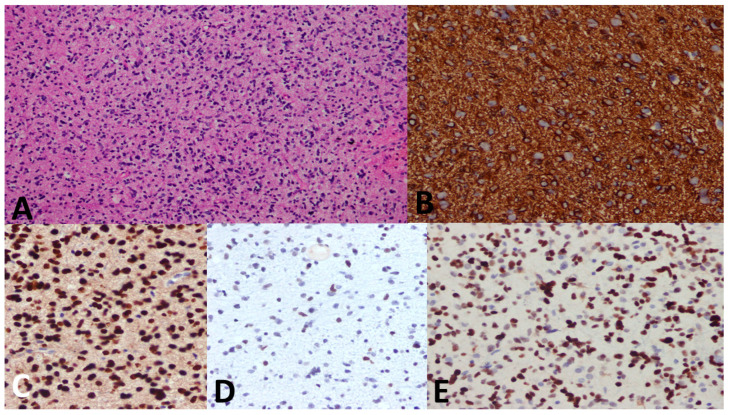
A 5-year-old female child with a pontine mass. (**A**) Biopsy shows an infiltrative and highly cellular tumor with pleomorphism and mitosis. (**B**) GFAP immunostain is diffusely positive. (**C**) H3 K27M immunostain shows strong nuclear stain in the tumor cells denoting mutation (endothelium serves as the internal negative control). (**D**) H3 K27me3 immunostain shows attenuated immunostain (strong nuclear stain is seen in normal cells). (**E**) p53 immunostain shows strong diffuse nuclear staining. Diagnosis was established as DMG, H3 K27–altered, CNS WHO grade 4.

**Figure 7 cancers-15-04545-f007:**
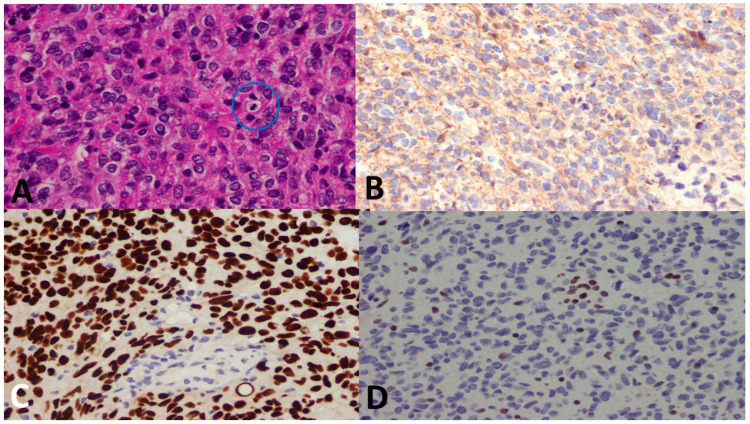
A 14-year-old female patient with a spinal cord tumor. (**A**) There is a highly cellular tumor with mitosis (blue circle). (**B**) GFAP immunostain was negative (Oligo-2 was positive, not shown). (**C**) H3 K27M shows a strong nuclear stain (endothelium serves as the internal negative control), whereas (**D**) H3 K27me3 immunostain shows loss of nuclear stain (endothelium serves as the internal positive control). The final diagnosis was DMG, H3 K27–altered, CNS WHO grade 4.

**Table 1 cancers-15-04545-t001:** Demographics, clinical, and immunohistochemical data stratified by the primary site of the tumor.

	Total (n = 104) n, %	Thalamic (n = 31) n, %	Brainstem (n = 13) n, %	Spinal Cord (n = 23) n, %	Basal Ganglia (n = 10) n, %	Corpus Callosum (n = 22) n, %	Other (n = 5) n, %	*p*-Value
**Median age in years (IQR)**	23 (10–48)	18 (3–69)	9 (2–63)	8 (1–48)	48 (6–60)	53 (20–74)	18 (15–56)	0.698
**Age Group**								**0.001**
Adults	56 (58.33%)	15 (48.38%)	5 (38.46%)	9 (39.13%)	7 (78%)	17 (100%)	3 (60%)	
Pediatrics	40 (41.67%)	14 (51.62%)	8 (61.54%)	14 (60.87%)	2 (22%)	0 (0%)	2 (40%)	
**Gender**								0.405
Male	64 (61.54%)	17 (54.84%)	7 (53.85%)	12 (52.17%)	8 (80%)	16 (72.73%)	4 (75%)	
Female	40 (38.46%)	14 (45.16%)	6 (46.15%)	11 (47.83%)	2 (20%)	6 (27.27%)	1 (25%)	
**Diagnostic Entity**								**0.000**
Circumscribed glioma	21 (20.19%)	4 (12.9%)	5 (38.46%)	10 (43.48%)	2 (20%)	0 (0%)	0 (0%)	
Diffuse midline glioma	30 (28.85%)	17 (54.84%)	5 (38.46%)	8 (34.78%)	0 (0%)	0 (0%)	0 (0%)	
Diffuse high-grade glioma	43 (41.35%)	8 (25.81%)	2 (15.39%)	3 (13.04%)	4 (40%)	21 (95.45%)	5 (100%)	
Diffuse low-grade glioma	10 (9.61%)	2 (6.45%)	1 (7.69%)	2 (8.7%)	4 (40%)	1 (4.55%)	0 (0%)	
**Immunohistochemistry**								
GFAP	74 (71.15%)	27 (87.1%)	13 (100%)	20 (86.96%)	8 (80%)	2 (9.1%)	4 (80%)	**0.000**
IDH1 (p.R132H)	2 (1.92%)	0 (0%)	0 (0%)	0 (0%)	2 (20%)	0 (0%)	0 (0%)	**0.002**
p53	26 (25%)	11 (35.48%)	4 (30.77%)	4 (17.39%)	5 (50%)	1 (4.55%)	1 (20%)	**0.050**
ATRX	12 (11.54%)	9 (29.03%)	1 (7.69%)	2 (8.7%)	0 (0%)	0 (0%)	0 (0%)	**0.013**
H3 K27M	21 (20.19%)	11 (35.48%)	4 (30.77%)	6 (26.09%)	0 (0%)	0 (0%)	0 (0%)	**0.009**
H3 K27me3	30 (28.85%)	17 (54.84%)	5 (38.46%)	8 (34.78%)	0 (0%)	0 (0%)	0 (0%)	**0.000**
**Intervention**								
Chemotherapy	44 (42.31%)	11 (35.48%)	3 (23.08%)	13 (56.52%)	8 (80%)	6 (27.27%)	3 (60%)	**0.024**
Radiotherapy	54 (51.92%)	14 (45.16%)	2 (15.39%)	12 (52.17%)	8 (80%)	15 (68.18%)	3 (60%)	**0.022**
**Outcome**								0.208
Alive	26 (25%)	7 (22.58%)	3 (23.08%)	10 (43.48%)	3 (30%)	1 (4.55%)	2 (40%)	
Dead	45 (43.27%)	16 (51.61%)	4 (30.77%)	8 (34.78%)	5 (50%)	10 (45.45%)	2 (40%)	
Lost to Follow-up	33 (31.73%)	7 (25.81%)	6 (46.15%)	5 (21.74%)	2 (20%)	11 (50%)	1 (20%)	

**Table 2 cancers-15-04545-t002:** Midline gliomas classified based on their diagnostic entity and immunohistochemical characteristics.

	Total (n = 104) n, %	Diffuse Midline Glioma (n = 30) n, %	Diffuse High-Grade Glioma (n = 43) n, %	Diffuse Low-Grade Glioma (n = 10) n, %	Circumscribed Glioma (n = 21) n, %	*p*-Value
**Median age in years (IQR)**	23 (10–48)	19.5 (3–67)	46 (2–74)	45.5 (2–59)	8 (1–48)	0.202
**Age group**						**0.001**
Adult	56 (58.33%)	17 (56.67%)	29 (80.55%)	5 (55.56%)	5 (23.81%)	
Pediatric	40 (41.67%)	13 (43.33%)	7 (19.45%)	4 (44.44%)	16 (76.19%)	
**Gender**						0.269
Male	64 (61.54%)	15 (50%)	29 (67.44%)	8 (80%)	12 (57.14%)	
Female	40 (38.46%)	15 (50%)	14 (32.56%)	2 (20%)	9 (42.86%)	
**Location**						**0.000**
Thalamic	31 (29.8%)	17 (56.67%)	8 (18.6%)	2 (20%)	4 (19.05%)	
Brainstem	13 (12.5%)	5 (16.66%)	2 (4.65%)	1 (10%)	5 (23.81%)	
Spinal Cord	23 (22.1%)	8 (26.67%)	3 (6.98%)	2 (20%)	10 (47.62%)	
Basal Ganglia	10 (9.6%)	0 (0%)	4 (9.4%)	4 (40%)	2 (9.52%)	
Corpus Callosum	22 (21.2%)	0 (0%)	21 (48.83%)	1 (10%)	0 (0%)	
Other	5 (4.8%)	0 (0%)	5 (11.63%)	0 (0%)	0 (0%)	
**Immunohistochemistry**						
GFAP	74 (71.15%)	28 (93.33%)	21 (48.84%)	6 (60%)	19 (90.48%)	**0.000**
IDH1	2 (1.92%)	0 (0%)	0 (0%)	2 (20%)	0 (0%)	**0.000**
p53	26 (25%)	15 (50%)	10 (23.26%)	1 (10%)	0 (0%)	**0.000**
ATRX	12 (11.54%)	10 (33.33%)	2 (4.65%)	0 (0%)	0 (0%)	**0.000**
H3 K27M	21 (20.19%)	21 (70%)	0 (0%)	0 (0%)	0 (0%)	**0.000**
H3 K27me3	30 (28.85%)	30 (100%)	0 (0%)	0 (0%)	0 (0%)	**0.000**
**Intervention**						
Chemotherapy	44 (42.31%)	11 (36.67%)	20 (46.51%)	5 (50%)	8 (38.1%)	0.778
Radiotherapy	54 (51.92%)	15 (50%)	29 (67.44%)	6 (60%)	4 (19.05%)	**0.004**
**Outcome**						**0.001**
Alive	26 (25%)	6 (20%)	7 (16.28%)	4 (40%)	9 (42.86%)	
Dead	45 (43.27%)	18 (60%)	23 (53.49%)	3 (30%)	1 (4.76%)	
Lost to Follow-up	33 (31.73%)	6 (20%)	13 (30.23%)	3 (30%)	11 (52.38%)	

**Table 3 cancers-15-04545-t003:** Predictors of overall survival among patients with midline gliomas.

	HR	*p*-Value	95% CI
**Age**	1.010	0.546	0.982–1.034
**Gender**			
Female	1.000	-	-
Male	0.673	0.339	0.299–1.514
**Location**			
Brainstem	1.000	-	-
Thalamus	1.210	0.777	0.330–4.407
Spinal Cord	1.470	0.621	0.318–6.793
Corpus Callosum	1.630	0.561	0.315–8.391
Basal Ganglia	2.870	0.276	0.429–19.218
Other	1.270	0.839	0.123–13.141
**Group**			
Diffuse high-grade glioma	1.000	-	-
Diffuse midline glioma	0.890	0.817	0.320–2.451
Diffuse low-grade glioma	0.310	0.103	0.075–1.265
Circumscribed glioma	0.013	**0.000**	0.001–0.142
**Intervention**			
Chemotherapy *	1.430	0.436	0.582–3.509
Radiotherapy *	0.300	**0.048**	0.090–0.987

(*): The reference is the absence of using the treatment method.

## Data Availability

The data presented in this study are available on request from the corresponding author.

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
