# Peer review of "Midline Gliomas: A Retrospective Study from a Cancer Center in the Middle East"

_cancers, 2023, doi:10.3390/cancers15184545_

Round 1
Reviewer 1 Report
In manuscript by Al Sharie et al., authors reported their findings of midline gliomas, both diffuse and circumscribed, conducted at King Hussein Cancer Center. This study is acceptable for publication, but it has a few limitations:
The main limitation of this study is the loss of follow up of nearly 32% of all patients. The authors acknowledged this limitation but it is almost one third of all patients and comparison the probability of survival between different demographic, clinical, and immunohistochemical characteristics are very restricted.
In section Results, line 157, authors stated that “There was a statistically significant difference in all three treatment options between the location groups of the study”. However, in Tables 1 and 2 we can see that patients received only two treatment options, radiotherapy and chemotherapy. We know that standard treatment protocol for high-grade gliomas include combination of radiotherapy and temozolomide therapy, but temozolomide improves patient survival by only 2.5 months compared to radiotherapy alone. Which chemotherapy regimen did patients receive in this study? Authors also stated that treatment options for DMG are confined to either chemotherapy, radiotherapy, or targeted molecular agents. Which targeted therapy is approved for DMG treatment?
DMG is often encountered in children with a median age of 9 years, representing 10.0% of all pediatric brain tumors. However, in this study the number of adults (n=17, 56.7%) diagnosed with DMGs outweighed that of children. This information should be presented in Results and table 2, not only in discussion.
In this study, most cases of DMGs were located in the thalamus (n=17, 56.67%), while the remaining occurred in the spinal cord (n=8, 26.66%) and brainstem (n=5, 16.67%). However, in Table 2 we can see that two additional cases are located in basal ganglia, two in corpus callosum and two in other structures (36 patients in total diagnosed with DMG, not 30).
Sentence in Introduction from line 44 to line 49 is too long and little bit confusing so authors should rephrase that.
Author Response
We thank the reviewer for their insightful comments and valid concerns. Here are the answers to the comments.
Comment:
In manuscript by Al Sharie et al., authors reported their findings of midline gliomas, both diffuse and circumscribed, conducted at King Hussein Cancer Center. This study is acceptable for publication, but it has a few limitations:
The main limitation of this study is the loss of follow up of nearly 32% of all patients. The authors acknowledged this limitation, but it is almost one third of all patients and comparison the probability of survival between different demographic, clinical, and immunohistochemical characteristics are very restricted.
Response:
Thank you for your careful review of our manuscript, we appreciate your thoughtful comments and suggestions. We agree with your assessment that the main limitation of our study is the loss of follow-up for nearly 32% of all patients. This is a significant limitation of our patient’s cohort in general mainly because some of our patients come from outside the country (around 25-30% of our patient population according to the Center’s Cancer Registry) and leave back to their home countries once they complete the diagnosis and treatment, leaving us with no appropriate follow-up. We have added this in the limitation section of the study in lines 411-417.
Comment:
In section Results, line 157, authors stated that “There was a statistically significant difference in all three treatment options between the location groups of the study”. However, in Tables 1 and 2 we can see that patients received only two treatment options, radiotherapy and chemotherapy.
Response:
Thank you for bringing this to our attention. It appears there was a mistake in our initial description. We apologize for any confusion caused. The sentence in question should have stated that "There was a statistically significant difference in both treatment options between the location groups of the study," as reflected in Tables 1 and 2 where patients received two treatment options, radiotherapy and chemotherapy. We appreciate your diligence in reviewing our work, and we have updated the correction in the results section.
Comment:
We know that standard treatment protocol for high-grade gliomas include combination of radiotherapy and temozolomide therapy, but temozolomide improves patient survival by only 2.5 months compared to radiotherapy alone. Which chemotherapy regimen did patients receive in this study? Authors also stated that treatment options for DMG are confined to either chemotherapy, radiotherapy, or targeted molecular agents. Which targeted therapy is approved for DMG treatment?
Response:
We thank the reviewer for their suggestion. Accordingly, we have added a sentence in the results section lines 179-181 stating that patients diagnosed with pontine gliomas were treated with focal radiotherapy with or without prednisolone. However, patients diagnosed with DMG at other locations including thalamic DMGs are treated with focal radiotherapy with daily temozolomide. Also, we have added a paragraph to the discussion section describing the use of targeted molecular therapy for DMGs in the discussion section in lines 400-410. However, targeted therapy is not used at our center for this group of patients.
Comment:
DMG is often encountered in children with a median age of 9 years, representing 10.0% of all pediatric brain tumors. However, in this study the number of adults (n=17, 56.7%) diagnosed with DMGs outweighed that of children. This information should be presented in Results and table 2, not only in discussion.
Response:
We thank you for your comment. Accordingly, this was added to the results section in lines 168-169 and lines 235-236 and the study population was subdivided into adults and pediatrics in tables 1 and 2. Please note that these numbers reflect the cases that underwent biopsy only, which might not be a real reflection of the total number, and thus percentage, of tumors in adults vs. pediatrics. So, patients that were diagnosed on the basis of MRI only were not included in the study. This point now has been clearly added to the methods as well in line 93
Comment:
In this study, most cases of DMGs were located in the thalamus (n=17, 56.67%), while the remaining occurred in the spinal cord (n=8, 26.66%) and brainstem (n=5, 16.67%). However, in Table 2 we can see that two additional cases are located in basal ganglia, two in corpus callosum and two in other structures (36 patients in total diagnosed with DMG, not 30).
Response:
Thank you for your observation and clarification. We sincerely appreciate your meticulous review of our study. Upon careful reevaluation, we acknowledge that there was an error in copying from our analysis sheet which led to the inclusion of cases in the basal ganglia, corpus callosum, and other structures. We apologize for any confusion this may have caused. Our study encompasses a total of 30 patients diagnosed with DMGs across the classical midline locations (thalamus, brainstem, and spinal cord), rather than the previously mentioned 36 cases. We have rechecked the analysis for all of the variables and have ensured that there were no other mistakes. Your diligence has contributed significantly to the accuracy and integrity of our findings, and we are grateful for your attention to detail. As correctly stated in our results and discussion, there were no DMG cases in locations outside the classical midline locations.
Comment:
Sentence in Introduction from line 44 to line 49 is too long and little bit confusing so authors should rephrase that.
Response:
Thank you for your comment. Accordingly, we have rephrased the mentioned sentence.
Reviewer 2 Report
The authors describe an interesting, rather large series of a rare tumor. Which is important to collect these data retrospectively. I do have a few comments.
Major comments
1. In the statistical analysis, you mention "Also, a Cox Hazard regression model was built with predictors chosen based on the literature." line 141. Please specify which factors you included. There are many factors that have been described in the literature, some more solid than others.
2. The Tables are missing a solid desription, please explain the abbreviations under the Table. In addition, what are the p-values reflection. The differences the locations or, the type of tumor, or ..... In a series of 104 patients and 5 subgroups, significant p-values are easy to obtain, what is the clinical relevance of these?
3. Subsequently, the results section is rather long, with a description of all the p-values and with the same information of the Tables, please summarize the most important findings and reflect on these findings, what is the clinical relevance of these.
A few minor comments:
line 47: "...which were managed at KHCC between...", I would change mange into diagnosed and treated at KHCC..."
Line 147: please mention the mean with a standard deviation or a median with a range or IQR, bot both.
Line 153: please rephrase, it is unclear in which subgroups the differences was not significant, Suggestion " Except in the DMG group (50% male and 50% female), there was a male predominance noticed in all groups, although this was not statistically significant (p = 0.269)."
Line 156: Please clarify, which patients received more chemotherapy and which more radiotherapy or perhaps both. I believe it is not sufficient just to mention Table 1 here. "Fifty-four (51.92%) patients received radiotherapy, and 44 (42.31%) received chemotherapy. There was a statistically significant difference in all three treatment options between the location groups of the study (p < 0.05)"
Apart from the comments above no addition comments regarding the quality of English language
Author Response
We thank the reviewer for their insightful comments and valid concerns. Here are the answers to the comments.
Comment:
The authors describe an interesting, rather large series of a rare tumor. Which is important to collect these data retrospectively. I do have a few comments.
Major comments
- In the statistical analysis, you mention "Also, a Cox Hazard regression model was built with predictors chosen based on the literature." line 141. Please specify which factors you included. There are many factors that have been described in the literature, some more solid than others.
Response:
Thank you for your comment. To ensure a comprehensive understanding of the variables associated with mortality in midline gliomas, we conducted a thorough literature review. Our objective was to identify the most relevant and substantiated factors that have consistently demonstrated significance in previous studies.
Following this rigorous review, we selected several key variables that have shown strong evidence of influence on mortality outcomes for midline gliomas, both in the broader context and specifically for diffuse midline gliomas (DMGs). These variables include age, gender, tumor location, WHO grade, and treatment method, in particular radiotherapy. We acknowledge the wide array of factors described in the literature.
Furthermore, in line with your observation, we employed a univariate regression model to assess the individual impact of these variables on mortality. Subsequently, the factors deemed significant in the univariate analysis were included in the Cox Hazard regression analysis to ascertain their collective influence while accounting for potential confounders. We have added the included variables and their methods of inclusion to the methods section in lines 152-158.
- The Tables are missing a solid description, please explain the abbreviations under the Table. In addition, what are the p-values reflection. The differences between the locations or, the type of tumor, or ..... In a series of 104 patients and 5 subgroups, significant p-values are easy to obtain, what is the clinical relevance of these?
Response:
Thank you for your comment. We acknowledge this comment, however, beyond the significance (or not) of the p-value, we thought that our study addresses the differences in the diagnosis and outcome of diffuse glioma and circumscribed gliomas in midline structures, whether in classical locations or in more recently described midline locations, based on recent literature. We have shown that in our relatively large cohort of patients in this rare tumor type, the locations and the staining patterns are consistent with what is described in the WHO and does not extend beyond the classical locations and histology.
- Subsequently, the results section is rather long, with a description of all the p-values and with the same information of the Tables, please summarize the most important findings and reflect on these findings, what is the clinical relevance of these.
Response:
Thank you for your comment, which we totally agree with. We tried to shorten the results as much as we thought would be appropriate so that results are addressed adequately.
A few minor comments:
- line 47: "...which were managed at KHCC between...", I would change mange into diagnosed and treated at KHCC..."
Response:
We thank you for your comment. Accordingly, we have changed the sentence as required.
- Line 147: please mention the mean with a standard deviation or a median with a range or IQR, bot both.
Response:
Thank you for your comment. Accordingly, the required edits were made.
- Line 153: please rephrase, it is unclear in which subgroups the differences was not significant, Suggestion " Except in the DMG group (50% male and 50% female), there was a male predominance noticed in all groups, although this was not statistically significant (p = 0.269)."
Response:
Thank you for your comment. Accordingly, the sentence was replaced with the suggested sentence.
- Line 156: Please clarify, which patients received more chemotherapy and which more radiotherapy or perhaps both. I believe it is not sufficient just to mention Table 1 here. "Fifty-four (51.92%) patients received radiotherapy, and 44 (42.31%) received chemotherapy. There was a statistically significant difference in all three treatment options between the location groups of the study (p < 0.05)"
Response:
We thank the reviewer for their important suggestions. Accordingly, we have further demonstrated the difference between tumor locations and their intake of chemotherapy/ radiotherapy in lines 174-178.
Reviewer 3 Report
The authors present essentially a descriptive study of an institutional series of gliomas affecting midline structures, at a large national cancer center serving 60% of Jordan. Findings corroborate prior series. Strengths of the study are the large center serving a significant population at a nation level. The series is of reasonable interest as a descriptive series of these tumors at a small population level from a non-US and non-European center. I have the following comments:
1) more details needed in methods about the statistical analyses. How were variables chosen for Cox? Why? How do the variables appear in univariate analysis? Multivariate?
2) Radiation was described as associated with survival. This should be explored further if it is a main conclusion. Authors should offer characteristics of patients broken down by receipt of radiation or not. It would be nice to see KM curves for this breakdown. Is there selection bias?
3) report numbers to a consistent number of significant digits.
4) please provide more information about treatments. Surely, patients did not simply receive either chemotherapy or radiation, but many received both in varying sequences? which chemotherapies were used?
Minor edits to grammar throughout
Author Response
We thank the reviewer for their insightful comments and valid concerns. Here are the answers to the comments.
Comment:
The authors present essentially a descriptive study of an institutional series of gliomas affecting midline structures, at a large national cancer center serving 60% of Jordan. Findings corroborate prior series. Strengths of the study are the large center serving a significant population at a nation level. The series is of reasonable interest as a descriptive series of these tumors at a small population level from a non-US and non-European center. I have the following comments:
1) more details needed in methods about the statistical analyses. How were variables chosen for Cox? Why?
Response:
Thank you for your comment. To ensure a comprehensive understanding of the variables associated with mortality in midline gliomas, we conducted a thorough literature review. Our objective was to identify the most relevant and substantiated factors that have consistently demonstrated significance in previous studies.
Following this rigorous review, we selected several key variables that have shown strong evidence of influence on mortality outcomes for midline gliomas, both in the broader context and specifically for diffuse midline gliomas (DMGs). These variables include age, gender, tumor location, WHO grading, and treatment method. We acknowledge the wide array of factors described in the literature.
Furthermore, in line with your observation, we employed a univariate regression model to assess the individual impact of these variables on mortality. Subsequently, the factors deemed significant in the univariate analysis were included in the Cox Hazard regression analysis to ascertain their collective influence while accounting for potential confounders. We have added the included variables and their methods of inclusion to the methods section in lines 152-158.
2) Radiation was described as associated with survival. This should be explored further if it is a main conclusion. Authors should offer characteristics of patients broken down by receipt of radiation or not. It would be nice to see KM curves for this breakdown. Is there selection bias?
Response:
Thank you for your comment. Accordingly, we have added a new KM figure for the treatment methods for our midline glioma cases (chemotherapy and radiothereapy). The results of which were added to the results section in lines 295-300. Moreover, our sample of participants was selected from a list of all patients who were diagnosed with midline gliomas at our center. We did not exclude any participants based on their age, sex, race, ethnicity, or socioeconomic status. We believe that our sample of participants is representative of the overall population of patients with midline gliomas. And when compared to the literature, we are confident that our findings are not due to selection bias.
3) report numbers to a consistent number of significant digits.
Response:
Thank you for your comment. Accordingly, the manuscript was rechecked, and we made sure to report the numbers to a consistent number of digits.
4) please provide more information about treatments. Surely, patients did not simply receive either chemotherapy or radiation, but many received both in varying sequences? which chemotherapies were used?
Response:
Thank you for your comment. Accordingly, we have added a sentence in the results section lines 179-181 stating that patients diagnosed with pontine gliomas were treated with focal radiotherapy with or without cortisol. However, patients diagnosed with DMG at other locations including thalamic DMGs are treated with focal radiotherapy with daily temozolomide.
Round 2
Reviewer 1 Report
Authors accepted all my suggestions and manuscript is acceptable for publication in present form.
Author Response
Comment:
Authors accepted all my suggestions, and the manuscript is acceptable for publication in present form.
Response:
Thank you for your kind words. We are very pleased that you found our manuscript acceptable for publication in its present form. We have carefully considered all of your suggestions and made the necessary revisions. We believe that the manuscript is now stronger and more rigorous as a result of your feedback.
Reviewer 3 Report
Thank you for addressing the comments and for the edits to the manuscript. I have a few remaining suggestions.
1) Please edit the grammar throughout. In some areas with new edits, there are errors in English.
2) It would be helpful to describe in more detail the treatment practices described -- you report that spinal and thalamic tumors were treated more frequently with chemotherapy -- why?
3) I asked previously about what salvage treatments, if any, patients received. It would be helpful even if described qualitatively -- e.g describing what the typical salvage treatment practice is, or if no salvage treatment was typically given in favor of supportive care.
4) Please describe in the limitations the fact that on kaplan meier analysis, radiotherapy is associated with numerically worse outcome, although this was not statistically significant. This numerical trend is reversed when the authors perform a Cox regression in favor of radiotherapy, which is unusual and thus the limitations should describe that this finding is limited by the small sample size and retrospective nature of this work.
Minor edits are needed. There are some grammar error introduced around some of the revisions the authors made.
Author Response
Comment-1:
Please edit the grammar throughout. In some areas with new edits, there are errors in English.
Response:
Thank you for your comment. We have checked the grammar throughout the manuscript and have corrected it in the new edits.
Comment-2:
It would be helpful to describe in more detail the treatment practices described -- you report that spinal and thalamic tumors were treated more frequently with chemotherapy -- why?
Response:
Thank you for your comment. Per the Cancer Registry from KHCC, we were provided with a brief description of the regimens used in lines 178-181. Chemotherapy consisted primarily of Temozomolide, most commonly as the only agent, with occasional cases receiving Lomustine in the pediatric age group. Vincristine with carboplatin was the second most common regimen received, mostly in pediatric age group. Radiotherapy was received mostly at a dose of 5400 Gys. As for the thalamic and spinal cord tumors, please note that the patients at our center are treated by 2 different teams according to the age of the patient, pediatric (younger than 18 years of age) and adults (18 years of age and older). There are differences in the approach of the treatment between adult and pediatric patients’ teams, which is reflected on the differences in the treatment approaches.
Comment-3:
I asked previously about what salvage treatments, if any, patients received. It would be helpful even if described qualitatively -- e.g describing what the typical salvage treatment practice is, or if no salvage treatment was typically given in favor of supportive care.
Response: Thank you for your comment. For DMG, no salvage treatment are given beyond the initial treatment, and these are typically offered supportive care. Whereas patients with low grade gliomas are offered several lines of chemotherapy once they progress on the first line of treatment. As per the comment above, we have 2 teams that look after neuro-oncology: the pediatric and adult teams. Whereas the neurosurgeons, radiologists, radiation oncologists, and neuropathologists serve on both teams, the medical/ pediatric oncologists are different with different approaches to treatment which makes it hard to draw definite conclusions.
Comment-4:
Please describe in the limitations the fact that on Kaplan Meier analysis, radiotherapy is associated with numerically worse outcome, although this was not statistically significant. This numerical trend is reversed when the authors perform a Cox regression in favor of radiotherapy, which is unusual and thus the limitations should describe that this finding is limited by the small sample size and retrospective nature of this work.”
Response:
We thank you for your comment. Accordingly, we have added this to the limitations section of our study lines 416-420.